# LARGE-SCALE CLOZE TEST DATASET DESIGNED BY TEACHERS

## ABSTRACT

Cloze test is widely adopted in language exams to evaluate students' language proficiency. In this paper, we propose the first large-scale human-designed cloze test dataset CLOTH [1], in which the questions were used in middle-school and high-school language exams. With the missing blanks carefully created by teachers and candidate choices purposely designed to be confusing, CLOTH requires a deeper language understanding and a wider attention span than previous automatically generated cloze datasets. We show humans outperform dedicated designed baseline models by a significant margin, even when the model is trained on sufficiently large external data. We investigate the source of the performance gap, trace model deficiencies to some distinct properties of CLOTH, and identify the limited ability of comprehending a long-term context to be the key bottleneck. In addition, we find that human-designed data leads to a larger gap between the model's performance and human performance when compared to automatically generated data.

## 1 INTRODUCTION

Being a classic language exercise, the cloze test (Taylor, 1953) is an accurate assessment of language proficiency (Fotos, 1991; Jonz, 1991; Tremblay, 2011) and has been widely employed in language examinations. Under standard setting, a cloze test requires examinees to fill in the missing word (or sentence) that best fits the surrounding context. To facilitate natural language understanding, automatically generated cloze datasets were introduced to measure the ability of machines in reading comprehension (Hermann et al., 2015; Hill et al., 2016; Onishi et al., 2016). In these datasets, each cloze question typically consists of a context paragraph and a question sentence. By randomly replacing a particular word in the question sentence with a blank symbol, a single test case is created. For instance, the CNN/Daily Mail (Hermann et al., 2015) take news articles as the context and the summary bullet points as the question sentence. Only named entities are considered when creating the blanks. Similarly, in Children's Books test (CBT) (Hill et al., 2016), the cloze question is obtained by removing a word in the last sentence of every consecutive 21 sentences, with the first 20 sentences being the context. Different from the CNN/Daily Mail datasets, CBT also provides each question with a candidate answer set, consisting of randomly sampled words with the same part-of-speech tag from the context as that of the ground truth.

Thanks to the automatic generation process, these datasets can be very large in size, leading to significant research progress. However, compared to how humans would create cloze questions, the automatic generation process bears some inevitable issues. Firstly, the blanks are chosen uniformly without considering which aspect of the language phenomenon the question will test. Hence, quite a portion of automatically generated questions can be purposeless or even trivial to answer. Another issue involves the ambiguity of the answer. Given a context and a blanked sentence, there can be multiple words that fit almost equally well into the blank. A possible solution is to include a candidate option set, as done by CBT, to get rid of the ambiguity. However, automatically generating the candidate option set can be problematic since it cannot guarantee the ambiguity is removed. More importantly, automatically generated candidates can be totally irrelevant or simply grammatically unsuitable for the blank, resulting in again trivial questions. Probably due to these unsatisfactory issues, it has been shown neural models have achieved comparable performance with human within

---

[1]CLOTH (**CLO**ze test by **T**eac**H**ers) will be made public.

very short time (Chen et al., 2016; Dhingra et al., 2016; Seo et al., 2016). While there has been work trying to incorporate human design into cloze question generation (Zweig & Burges, 2011), the MSR Sentence Completion Challenge created by this effort is quite small in size, limiting the possibility of developing powerful neural models on it.

Motivated by the aforementioned drawbacks, we propose CLOTH, a large-scale cloze test dataset collected from English exams. Questions in the dataset are designed by middle-school and high-school teachers to prepare Chinese students for entrance exams. To design a cloze test, teachers firstly determine the words that can test students' knowledge of vocabulary, reasoning or grammar; then replace those words with blanks and provide three candidate options for each blank. If a question does not specifically test grammar usage, all of the candidate options would complete the sentence with correct grammar, leading to highly confusing questions. As a result, human-designed questions are usually harder and are a better assessment of language proficiency. Note that, different from the reading comprehension task, a general cloze test does not focus on testing reasoning abilities but evaluates several aspects of language proficiency including vocabulary, reasoning and grammar.

To verify if human-designed cloze questions are difficult for current models, we train dedicated models as well as the state-of-the-art language model and evaluate their performance on this dataset. We find that the state-of-the-art model lags behind human performance even if the model is trained on a large external corpus. We analyze where the model fails compared to human. After conducting error analysis, we assume the performance gap results from the model's inability to use long-term context. To verify this assumption, we evaluate humans' performance when they are only allowed to see one sentence as the context. Our assumption is confirmed by the matched performances of the model and human when given only one sentence. In addition, we demonstrate that human-designed data is more informative and more difficult than automatically generated data. Specifically, when the same amount of training data is given, human-designed training data leads to better performance. Additionally, it is much easier for the same model to perform well on automatically generated data.

## 2 CLOTH DATASET

In this section, we introduce the CLOTH dataset that is collected from English examinations, and study the assessed abilities of this dataset.

### 2.1 DATA COLLECTION AND STATISTICS

We collected the raw data from three free websites[2] in China that gather exams designed by English teachers. These exams are used to prepare students for college/high school entrance exams. Before cleaning, there are $20,605$ passages and $332,755$ questions. We perform the following processes to ensure the validity of the data: 1. We remove questions with an inconsistent format such as questions with more than four options; 2. We filter all questions whose validity relies on external information such as pictures or tables; 3. Further, we delete duplicated passages; 4. On one of the websites, the answers are stored as images. We use two OCR software, tesseract[3] and ABBYY FineReader[4], to extract the answers from images. We discard the question when results from the two software are different. After the cleaning process, we obtain a dataset of $7,131$ passages and $99,433$ questions.

Since high school questions are more difficult than middle school questions, we divided the datasets into CLOTH-M and CLOTH-H, which stand for the middle school part and the high school part. We split $11\%$ of the data for both the test set and the dev set. The detailed statistics of the whole dataset and two subsets are presented in Table 1.

### 2.2 QUESTION TYPE ANALYSIS

In order to evaluate students' mastery of a language, teachers usually design tests so that questions cover different aspects of a language. Specifically, they first identity words in the passage that can

---

[2] http://www.21cnjy.com/; http://5utk.ks5u.com/; http://zujuan.xkw.com/

[3] https://github.com/tesseract-ocr

[4] https://www.abbyy.com/en-us/finereader/

| Dataset | CLOTH-M | | | CLOTH-H | | | CLOTH | | |
|---|---|---|---|---|---|---|---|---|---|
| Subset | Train | Dev | Test | Train | Dev | Test | Train | Dev | Test |
| # passages | 2,341 | 355 | 335 | 3,172 | 450 | 478 | 5,513 | 805 | 813 |
| # questions | 22,056 | 3,273 | 3,198 | 54,794 | 7,794 | 8,138 | 76,850 | 11,067 | 11,516 |
| # sentence | 16.26 | | | 18.92 | | | 17.79 | | |
| # words | 242.88 | | | 365.1 | | | 313.16 | | |
| Vocabulary size | 15,096 | | | 32,212 | | | 37,235 | | |

Table 1: The statistics of the training, dev and test sets of CLOTH-M (middle school questions), CLOTH-H (high school questions) and CLOTH

examine students knowledge in vocabulary, logic or grammar. Then, they replace the words with blanks and prepare three incorrect but confusing candidate options to make the test non-trivial. A sample passage is presented in Table 2.

---

**Passage:** Nancy had just got a job as a secretary in a company. Monday was the first day she went to work, so she was very _1_ and arrived early.

She _2_ the door open and found nobody there. "I am the _3_ to arrive." She thought and came to her desk. She was surprised to find a bunch of _4_ on it. They were fresh. She _5_ them and they were sweet. She looked around for a _6_ to put them in. "Somebody has sent me flowers the very first day!" she thought _7_ . " But who could it be?" she began to _8_ .

The day passed quickly and Nancy did everything with _9_ interest. For the following days of the _10_ , the first thing Nancy did was to change water for the followers and then set about her work.

Then came another Monday. _11_ she came near her desk she was overjoyed to see a(n) _12_ bunch of flowers there. She quickly put them in the vase, _13_ the old ones. The same thing happened again the next Monday. Nancy began to think of ways to find out the _14_ .

On Tuesday afternoon, she was sent to hand in a plan to the _15_ . She waited for his directives at his secretary's _16_ . She happened to see on the desk a half-opened notebook, which _17_ : "In order to keep the secretaries in high spirits, the company has decided that every Monday morning a bunch of fresh flowers should be put on each secretarys desk." Later, she was told that their general manager was a business management psychologist.

**Questions:**

| 1. | A. depressed | B. encouraged | **C. excited** | D. surprised |
|---|---|---|---|---|
| 2. | A. turned | **B. pushed** | C. knocked | D. forced |
| 3. | A. last | B. second | C. third | **D. first** |
| 4. | A. keys | B. grapes | **C. flowers** | D. bananas |
| 5. | **A. smelled** | B. ate | C. took | D. held |
| 6. | **A. vase** | B. room | C. glass | D. bottle |
| 7. | A. angrily | B. quietly | C. strangely | **D. happily** |
| 8. | A. seek | **B. wonder** | C. work | D. ask |
| 9. | A. low | B. little | **C. great** | D. general |
| 10. | A. month | B. period | C. year | **D. week** |
| 11. | A. Unless | **B. When** | C. Since | D. Before |
| 12. | A. old | B. red | C. blue | **D. new** |
| 13. | A. covering | B. demanding | **C. replacing** | D. forbidding |
| 14. | **A. sender** | B. receiver | C. secretary | D. waiter |
| 15. | A. assistant | B. colleague | C. employee | **D. manager** |
| 16. | A. notebook | **B. desk** | C. office | D. house |
| 17. | **A. said** | B. written | C. printed | D. signed |

Table 2: A Sample passage from our dataset. The correct answers are highlighted.

To understand the assessed abilities on this dataset, we divide questions into several types and label the proportion of each type of questions. We find that the questions can be divided into the following types:

- Grammar: The question is about grammar usage, involving tense, preposition usage, active/passive voices, subjunctive mood and so on.

- Short-term-reasoning: The question is about content words and can be answered based on the information within the same sentence.

- Matching/paraphrasing: The question is answered by copying/paraphrasing a word.

- Long-term-reasoning: The answer must be inferred from synthesizing information distributed across multiple sentences.

We sample 100 passages in the high school category and the middle school category respectively. Each passage in the high school category has 20 questions and each passage in the middle school category has 10 questions. The types of the 3000 question are labeled on Amazon Turk. We pay \$1 and \$0.5 for high school passage and middle school passage respectively.

The proportion of different questions is shown in Table 3. We find that the majority of questions are short-term-reasoning questions, in which the examinee needs to utilize grammar knowledge, vocabulary knowledge and simple reasoning to answer the questions. Note that questions in middle school are easier since they have more grammar questions. Finally, only approximately $22.4\%$ of data needs long-term information, in which the long-term-reasoning questions constitute a large proportion.

| Dataset | Short-term questions | | Long-term questions | | |
|---|---|---|---|---|---|
| | Grammar | Short-term-reasoning | Matching/paraphrasing | Long-term-reasoning | Others |
| CLOTH | 0.265 | 0.503 | 0.044 | 0.180 | 0.007 |
| CLOTH-M | 0.330 | 0.413 | 0.068 | 0.174 | 0.014 |
| CLOTH-H | 0.240 | 0.539 | 0.035 | 0.183 | 0.004 |

Table 3: The question type statistics of 3000 sampled questions. Grammar and short-term-reasoning questions can both be solved with a short context, while we need longer context to solve long-term-reasoning and matching/paraphrasing.

## 3 EXPLORING MODELS' LIMITS

In this section, we study if human-designed cloze test is a challenging problem for state-of-the-art models. We find that the language model trained on large enough external corpus could not solve the cloze test. After conducting error analysis, we hypothesize that the model is not able to deal with long-term dependencies. We verify the hypothesis by evaluating human's performance when human only see one sentence as the context.

### 3.1 HUMAN AND MODEL PERFORMANCE

**LSTM**  To test the performance of RNN based supervised models, we train a bidirectional LSTM (Hochreiter & Schmidhuber, 1997) to predict the missing word given the context, with only labeled data. The implementation details are in Appendix A.1.

**Attention Readers**  To enable the model to gather information from a longer context, we augment the supervised LSTM model with the attention mechanism (Bahdanau et al., 2014), so that the representation at the blank is used as a query to find the relevant context in the document and a blank-specific representation of the document is used to score each candidate answer. Specifically, we adapt the Stanford Attention Reader (Chen et al., 2016) and the position-aware attention model (Zhang et al., 2017) to the cloze test problem. With the position-aware attention model, the attention scores are based on both the context match and the distances of two words. Both attention models are trained only with the human-designed blanks just as the LSTM model.

**Language model**  Language modeling and cloze test are similar since, in both tasks, a word is predicted based on the context. In cloze test, the context on both sides may determine the correct answer. Suppose $x_i$ is the missing word and $x_1, \cdots, x_{i-1}, x_{i+1}, \cdots, x_n$ are the context. Although language model is trained to predict the next word only using the left context, to utilize the surrounding context, we could choose $x_i$ that maximizes the joint probability $p(x_1, \cdots, x_n)$, which essentially maximizes the conditional likelihood $p(x_{i-1} \mid x_1, \cdots, x_{i-1}, x_i, \cdots, x_n)$. Therefore, language model can be naturally adapted to cloze test.

In essence, language model treats each word as a possible blank and learns to predict it. As a result, it receives more supervision than the supervised model trained on human-labeled questions.

Additionally, it can be trained on a very large unlabeled corpus. Interested in whether the state-of-the-art language model can solve cloze test, we first train a neural language model on the training set of our corpus, then we test the language model trained on One Billion Word Benchmark (Chelba et al., 2013) (referred as 1-billion-language-model) that achieves a perplexity of 30.0 (Jozefowicz et al., 2016)[5]. To make the evaluation time tractable, we limit the context length to one sentence or three sentences.

**Human performance**  We measure the performance of Amazon Turkers on $3,000$ sampled questions when the whole passage is given.

The comparison is shown in Table 4. Both attention models achieve a similar accuracy to the LSTM. We hypothesize the attention model's unsatisfactory performance is due to the difficulty to learn to comprehend longer context when the majority of the training data only requires understanding short-term information. The language model trained on our dataset achieves an accuracy of $0.548$ while the supervised model's accuracy is $0.484$, indicating that more training data results in better generalization. When only one sentence is given as context, the accuracy of 1-billion-language-model is $0.695$, which shows that the amount of data is an essential factor affecting the model's performance. It also indicates that the language model can learn sophisticated language regularities when given enough data. The same conclusion can also be drawn from state-of-the-art results on six language tasks resulted from applying language model representations as word vectors (Anonymous, 2018). However, if we increase the context length to three sentences, the accuracy of 1-billion-language-model only improves to $0.707$. In contrast, human outperforms 1-billion-language-model by a significant margin, which demonstrates that deliberately designed questions in CLOTH are not completely solved even for state-of-the-art models.

| Model | CLOTH | CLOTH-M | CLOTH-H |
|---|---|---|---|
| LSTM | 0.484 | 0.518 | 0.471 |
| Stanford Attention Reader (Chen et al., 2016) | 0.487 | 0.529 | 0.471 |
| Position-aware Attention Reader | 0.485 | 0.523 | 0.471 |
| language model | 0.548 | 0.646 | 0.506 |
| 1-billion-language-model (one sentence) | 0.695 | 0.723 | 0.685 |
| 1-billion-language-model (three sentences) | 0.707 | 0.745 | 0.693 |
| human performance | 0.860 | 0.897 | 0.845 |

Table 4: Model and human's performance on CLOTH. Attention model does not leads to performance improvement compared to vanilla LSTM. Language model outperforms LSTM since it receives more supervisions in learning to predict each word. Training on large external corpus further significantly enhances the accuracy.

## 3.2 Analyzing model's performance by human study

In this section, we would like to understand why the state-of-the-art model lags behind human performance.

We find that most of the errors made by the large language model involve long-term reasoning. Additionally, in a lot of cases, the dependency is within the context of three sentences. Several errors made by the large language model are shown in Table 5. In the first example, the model does not know that Nancy found nobody in the company means that Nancy was the first one to arrive at the company. In the second and third example, the model fails probably because of the coreference from "they" to "flowers". The dependency in the last case is longer. It depends on the fact that "Nancy" was alone in the company.

Based on the case study, we hypothesize that the language model is not able to take long-term information into account, although it achieves a surprisingly good overall performance. Moreover, the 1-billion-language-model is trained on the sentence level, which might also result in paying more attention to short-term information. However, we do not have enough computational resources to train a large model on 1 Billion Word Benchmark to investigate the differences of training on sentence level or on paragraph level.

---

[5]The pre-trained model is obtained from https://github.com/tensorflow/models/tree/master/research/lm_1b

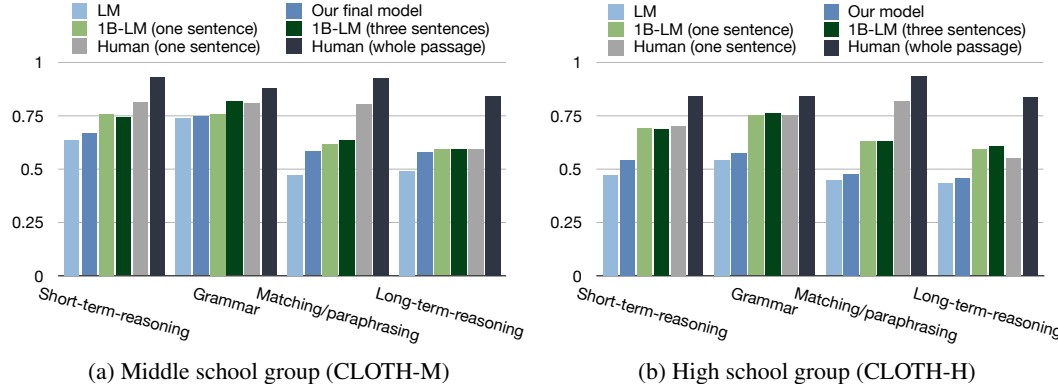

(a) Middle school group (CLOTH-M)  (b) High school group (CLOTH-H)

Figure 1: Model and human's performance on questions with different types. LM and 1B-LM denote language model trained on our dataset and the 1-Billion-Word corpus respectively. Our final model is introduced in Section 4.2. Training on large external corpus leads to improvements on all categories, showing that a large amount of data leads to substantial improvement in learning complex language regularities. When the human only has access to the context of one sentence, 1-billion-language-model is close to human's performance on most categories. Note that the accuracies on single categories may have high variance because of the relative small size of samples in each category.

| Context | Options | | | |
|---|---|---|---|---|
| She pushed the door open and found nobody there. "I am the __ to arrive." She thought and came to her desk. | *A. last* | B. second | C. third | **D. first** |
| They were fresh. She __ them and they were sweet. She looked around for a vase to put them in. | **A. smelled** | *B. ate* | C. took | D. held |
| She smelled them and they were sweet. She looked around for a __ to put them in. "Somebody has sent me flowers the very first day!" | **A. vase** | *B. room* | C. glass | D. bottle |
| "But who could it be?" she began to __ . The day passed quickly and Nancy did everything with great interest. | A. seek | **B. wonder** | C. work | *D. ask* |

Table 5: Error analysis of 1-billion-language-model with three sentences as the context. The questions are sampled from the sample passage shown in Table 2. The correct answer is in bold text. The incorrectly selected options are in italics.

An available comparison is to test the model's performance on different types of questions. We find that the model's accuracy is $0.591$ on long-term-reasoning questions of CLOTH-H while achieving $0.693$ on short-term-reasoning, which partially confirms that long-term-reasoning is harder. However, we could not completely rely on the performance on specific questions types, partly due to the small sample size. A more fundamental reason is that the question type labels are subjective and their reliability depends on whether turkers are careful enough. For example, in the error analysis shown in Table 5, a careless turker would label the second example as short-term-reasoning without noticing that the meaning of "they" relies on a long context span.

To objectively verify if the language model's strengths are in dealing with short-term information, we obtain the ceiling performance of only utilizing short-term information. Showing only one sentence as the context, we ask the turkers to label all possible options that they deem to be correct given the insufficient information. We also ask them to select a single option based on their best guesses. By limiting the context span manually, the ceiling performance with only the access to short context is estimated accurately.

The performances of turkers and 1-billion-language-model are shown in Table 6. The performance of 1-billion-language-model using one sentence as the context can almost match the ceiling performance of only using short-term information. Hence we conclude that the language model can almost perfectly solve all short-term cloze questions. However, the performance of language model is not improved significantly when the needed long-term context is given, indicating that the performance gap is due to the inability of long-term reasoning.

| Model | CLOTH | CLOTH-M | CLOTH-H |
|---|---|---|---|
| 1-billion-language-model (one sentence) | 0.695 | 0.723 | 0.685 |
| 1-billion-language-model (three sentences) | 0.707 | 0.745 | 0.693 |
| human (one sentence) | 0.714 | 0.771 | 0.691 |
| human (whole passage) | 0.860 | 0.897 | 0.845 |

Table 6: Human's performance compared with 1-billion-language-model

Assuming the majority of question type labels is reliable, we verify the strengths and weaknesses of models and human by studying the performance of models and human on different question categories. The comparison is shown in Figure 1.

The human study on short-term ceiling performance also reveals that the options are carefully picked. Specifically, when a Turker thinks that a question has multiple answers, $3.41$ out of $4$ options are deemed to be possibly correct, which means that teachers design the options so that three or four options all make sense if we only look at the local context.

## 4 COMPARING HUMAN-DESIGNED DATA AND AUTOMATICALLY GENERATED DATA

In this section, we demonstrate that human-designed data is a better test bed than automatically generated data for general cloze test since it results in a larger gap between the model's performance and human performance. However, the distributional mismatch between two types of data makes the human-designed data an unsuitable training source for solving automatically generated questions. In addition, we improve the model's performance by finding generated data that resembles human-designed data.

### 4.1 DATA COMPARISON

At a casual observation, a cloze test can be created by randomly deleting words and randomly sampling candidate options. In fact, to generate large-scale data, similar generation processes have been introduced and widely used in machine comprehension (Hermann et al., 2015; Hill et al., 2016; Onishi et al., 2016). However, research on cloze test design (Sachs et al., 1997) shows that tests created by deliberately deleting words are more reliable than tests created by randomly or periodically deleting words. To design accurate language proficiency assessment, teachers usually select words in order to examine students' proficiency in grammar, vocabulary and reasoning. Moreover, in order to make the question non-trivial, the three incorrect options provided by teachers are usually grammatically correct and relevant to the context. For instance, in the fourth problem of the sample passage shown in Table 2, "grapes", "flowers" and "bananas" all fit the description of being fresh. We know "flowers" is the correct answer after seeing the sentence "Somebody has sent me flowers the very first day!".

Naturally, we hypothesize that the distribution of human-generated data is different from automatically generated data. To verify this assumption, we compare the LSTM model's performance when given different proportion of the two types of data. Specifically, to train a model with $\alpha$ percent of automatically generated data, we randomly replace $a$ percent blanks with blanks at random positions, while keeping the remaining $100 - \alpha$ percent questions the same. The candidate options for the generated blanks are random words sampled from the unigram distribution. We test the trained model on human-designed data and automatically generated data respectively.

| Test Data | $\alpha = 0$ | $\alpha = 25$ | $\alpha = 50$ | $\alpha = 75$ | $\alpha = 100$ |
|---|---|---|---|---|---|
| Human-designed data | 0.484 | 0.475 | 0.469 | 0.423 | 0.381 |
| Automatically generated data | 0.422 | 0.699 | 0.757 | 0.785 | 0.815 |

Table 7: We train a model on $\alpha$ percent of automatically generated data and $100 - \alpha$ percent of human-designed data and test it on human-designed data and automatically generated data respectively.

The performance is shown in Table 7. We have the following observations: (1) human-designed data leads to a larger gap between the model's performance and the human performance, when given the same model. The model's performance and human's performance on the human-designed data are $0.484$ and $0.860$ respectively, leading to a gap of $0.376$. In comparison, the performance gap on the automatically generated data is at most $0.185$ since the model's performance reaches $0.815$ when trained on generated data. It shows that the distributions of human-designed data and automatically generated data are quite different. (2) the distributional mismatch between two types of data makes it difficult to transfer a model trained on human-designed data to automatically generated data. Specifically, the model's performance on automatically generated data monotonously increases when given a higher ratio of automatically generated training data.

To conclude, human-designed data is a good test base because of the larger gap between performances of the model and the human, although the distributional mismatch problem makes it difficult to be the best training source for out-of-domain cloze test such as automatically generated cloze test.

## 4.2 Combining Human-designed Data with Automatically Generated Data

In Section 3.1, we show that language model is able to take advantage of more supervisions since it predicts each word based on the context. In essence, each word can provide an automatically generated question. At the same time, we also show that human-designed data and the automatically generated data are quite different in Section 4.1. In this section, we propose to combine human-designed data with automatically generated data to achieve better performance.

Note that discriminative models can also treat all words in a passage as automatically generated questions, just like a language model (Please see the Appendix A.3 for details). We study two methods of leveraging automatically generated data and human-designed data:

**Equally averaging** Let $J_h$ be the average loss for all human-designed questions and $J_u$ be the average loss for all automatically generated questions in the passage. A straightforward method is to optimize $J_h + \lambda J_u$ so that the model learns to predict words deleted by human and all other words in the passage. We set $\lambda$ to 1 in our experiments. This model treats each automatically generated questions as equally important.

**Representativeness-based weighted averaging** A possible avenue towards having large-scale in-domain data is to automatically pick out questions which are representative of in-domain data among a large number of out-of-domain samples. Hence, we mimic the design behavior of language teachers by training a network to predict the representativeness of each automatically generated question. Note that the candidate option set for a automatically generated question is the whole vocabulary. We leave the candidate set prediction for future work. The performance of the representativeness prediction network and an example are shown in Appendix A.4.

Let $J_i$ denotes the negative log likelihood loss for the $i-$th question and let $l_i$ be the outputted representativeness of the $i$-th question (The definition of $l_i$ is in Appendix A.2). We define the representativeness weighted loss function as $J_f = \sum_{i \notin H} \text{Softmax}_i(\frac{l_1}{\alpha}, \cdots, \frac{l_n}{\alpha}) J_i$ where $H$ is the set of all human-generated questions and $\alpha$ is the temperature of the Softmax function. When the temperature is $+\infty$, the model degenerate into equally averaging objective function without using the representativeness. When the temperature is 0, only the most representative question is used. We set $\alpha$ to 2 based on the performance on the dev set.

We present the results in Table 8. When all other words are treated as equally important, the accuracy is $0.543$, similar to the performance of language model. Representativeness-based weighted averaging leads to an accuracy of $0.565$. When combined with human-designed data, the performance can be improved to $0.583$ [6].

## 5 Related Work

Large-scale automatically generated cloze test (Hermann et al., 2015; Hill et al., 2016; Onishi et al., 2016) leaded to significant research advancement. However, the generated questions do not consider

---

[6]The code will be available.

| Model | External Data | CLOTH | CLOTH-M | CLOTH-H |
|---|---|---|---|---|
| representativeness + human-designed ($J_f + J_h$) | | **0.583** | **0.673** | **0.549** |
| equal-average + human-designed ($J_u + J_h$) | | 0.566 | 0.662 | 0.528 |
| representativeness ($J_f$) | No | 0.565 | 0.665 | 0.526 |
| equal-average ($J_u$) | | 0.543 | 0.643 | 0.505 |
| human-designed ($J_h$) | | 0.484 | 0.518 | 0.471 |
| language model | | 0.548 | 0.646 | 0.506 |
| 1-billion-language-model (one sentence) | Yes | 0.695 | 0.723 | 0.685 |
| 1-billion-language-model (three sentences) | | *0.707* | *0.745* | *0.693* |
| Human (one sentence) | | 0.714 | 0.771 | 0.691 |
| Human (whole passage) | | 0.860 | 0.897 | 0.845 |

Table 8: Overall results on CLOTH. The "representativeness" means weighted averaging the loss of each question using the predicted representativeness. "equal-average" means to equally average losses of questions.

the language phenomenon to be tested and are relatively easy to solve. Recently proposed reading comprehension datasets are all labeled by human to ensure their qualities (Rajpurkar et al., 2016; Joshi et al., 2017; Trischler et al., 2016; Nguyen et al., 2016). Aiming to evaluate machines under the same conditions human is evaluated, there are a growing interests in obtaining data from examinations. NTCIR QA Lab (Shibuki et al., 2014) contains a set of real-world university entrance exam questions. The Entrance Exams task at CLEF QA Track (Peñas et al., 2014; Rodrigo et al., 2015) evaluates machine's reading comprehension ability. The AI2 Elementary School Science Questions dataset[7] provides 5,060 scientific questions used in elementary and middle schools. Lai et al. (2017) proposes the first large-scale machine comprehension dataset obtained from exams. They show that questions designed by teachers have a significant larger proportion of reasoning questions. Our dataset focuses on evaluating language proficiency while the focus of reading comprehension is reasoning.

In Section 4.2, we employ a simple supervised approach that predicts how likely a word is selected by teachers as a cloze question. It has been shown that features such as morphology information and readability are beneficial in cloze test prediction (Skory & Eskenazi, 2010; Correia et al., 2012; 2010). We leave investigating the advanced approaches of automatically designing cloze test to future work.

## 6 CONCLUSION

In this paper, we propose a large-scale cloze test dataset CLOTH that is designed by teachers. With the missing blanks and candidate options carefully created by teachers to test different aspects of language phenomenon, CLOTH requires a deep language understanding and better captures the complexity of human language. We find that human outperforms state-of-the-art models by a significant margin, even if the model is trained on a large corpus. After detailed analysis, we find that the performance gap is due to model's inability to understanding a long context. We also show that, compared to automatically-generated questions, human-designed questions are more difficult and leads to a larger margin between human performance and the model's performance.

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

years ago , if a teenager had some problems in his life , he might go home and write in his diary  now , a teenager with the same problems might go onto the internet and write about them in a blog . in many ways , a diary and a blog are very *similar* . but what makes UNK different from writing in a ( n ) *traditional* diary ? the biggest difference is that a blog is much more *public* than a diary . usually , a teenager treats his diary like a book full of *secrets* that he does not want to *share* with others . it's interesting that someone who writes in a blog instead of a diary will probably write nearly the same information . i have a little sister , and sometimes i go online to read her *blog* . she writes about things like waking up early for swimming practice and not studying enough for her chemistry test . *when* i was her age , i wrote about the same things , but *only* in my dairy . then , after i had finished writing , i would hide my diary in a secret place because i was *worried* that my sister might read it . the biggest *problem* with blogging is that anyone can read what you write . if i was angry with a friend during high school and wrote something *mean* about him in my diary , he would never know . *however* , if my sister ever wrote something bad about a friend , that friend might *read* her blog and get angry . there are also *advantages* to blogging , of course . if i was feeling sad one day and wrote in my diary , `` nobody cares about me '' , because no one would *know* about it . however , if my sister wrote the same sentence in her blog , her best friends would quickly *respond* and tell her how much they *like* her . blogs help people *stay* in contact with their friends and know what the people around them are doing .

Figure 2: Representativeness prediction for each word. Lighter color means less representative. The words deleted by human as blanks are in bold text.

## A    APPENDIX

### A.1    IMPLEMENTATION DETAILS

We implement our models using PyTorch[8]. The code of language model is adapted from the language model in PyTorch example projects[9]. We use Adam (Kingma & Ba, 2014) with the learning rate of 0.001. The hidden dimension is set to 650 and we initialize the word embedding by 300-dimensional Glove word vector (Pennington et al., 2014). The temperature $\alpha$ is set to 2. We train our model on all questions in CLOTH and test it on CLOTH-M and CLOTH-H separately.

### A.2    REPRESENTATIVENESS PREDICTION NETWORK

Let $x$ denote the passage and $z$ denote whether a word is selected as a question by human, i.e., $z$ is 1 if this word is selected to be filled in the original passage or 0 otherwise. Suppose $h_i$ is the representation of $i$-th word given by a bidirectional LSTM. The network computes the probability of $x_i$ being a question in the cloze test as follows:

$$l_i = h_i^T w_e; \quad p_i = \text{Sigmoid}(l_i)$$

where $l_i$ is the logit or the energy, indicating whether this word is likely to be a problem selected by human instructors. We train the network to minimize the binary cross entropy between $p$ and ground-truth labels at each token.

### A.3    USING ALL WORDS AS AUTOMATICALLY-GENERATED QUESTIONS

The passage $x$ is encoded by a bidirectional LSTM. Let $s_i$ be the context representation at $i$-th word. To mask the word $i$, $s_i$ is defined as $[\overrightarrow{h}_{i-1}, \overleftarrow{h}_{i+1}]^T$, where $\overrightarrow{h}_i$ and $\overleftarrow{h}_i$ are the hidden representations of forward LSTM and backward LSTM respectively. Suppose there are $m$ candidate words $w_{i,j}$ at position $i$, we denote the cross entropy loss $J_i$ between the model prediction $q_i$ and the ground-truth $y_i$ as,

$$q_i = \text{Softmax}(e_{w_{i,1}}^T W h_i, e_{w_{i,2}}^T W h_i, \cdots, e_{w_{i,m}}^T W h_i)$$
$$J_i = \text{Cross\_Entropy}(y_i, q_i)$$

where $e_{w_j}$ is the embedding of the word $w_j$ and $W$ is a weight matrix. In human-designed questions, there is a list of 4 candidate options for each question. For automatically-generated questions, the candidate options include the whole vocabulary, in which case $m$ is equal to the vocabulary size.

---

[8]http://pytorch.org/
[9]https://github.com/pytorch/examples/tree/master/word_language_model

## A.4 PERFORMANCE OF THE REPRESENTATIVENESS PREDICTION NETWORK

A predicted sample is shown in Figure 2. Clearly, words that are too obvious have low scores, such as punctuation marks, simple words "a" and "the". In contrast, content words whose semantics are directly related to the context have a higher score, e.g., "same", "similar", "difference" have a high score when the difference between two objects is discussed and "secrets" has a high score since it is related to the subsequent sentence "does not want to share with others".

Our prediction model achieves an F1 score of 36.5 on the test set, which is understandable since there are many plausible questions within a passage.

