# OpenReview forum: "Large-scale Cloze Test Dataset Designed by Teachers"
_ICLR.cc/2018/Conference — Reject_

### Official Review · AnonReviewer3 · 2017-11-15
**Promising dataset; needs better experiments to analyze**

**Rating:** 7
**Confidence:** 4

**Review:**

This paper presents a new dataset for cloze style question-answering. The paper starts with a very valid premise that many of the automatically generated cloze datasets for testing reading comprehension suffer from many shortcomings. The paper collects data from a novel source: reading comprehension data for English exams in China. The authors collect data for middle school and high school exams and clean it to obtain passages and corresponding questions and candidate answers for each question.

The rest of the paper is about analyzing this data and performance of various models on this dataset.

1) The authors divide the questions into various types based on the type of reasoning needed to answer the question, noticeably short-term reasoning and long-term reasoning.
2) The authors then show that human performance on this dataset is much higher than the performance of LSTM-based and language model-based baselines; this is in contrast to existing cloze style datasets where neural models achieve close to human performance.
3) The authors hypothesize that this is partially explained by the fact that neural models do not make use of long-distance information. The authors verify their claim by running human eval where they show annotators only 1 sentence near the empty slot and find that the human performance is basically matched by a language model trained on 1 billion words. This part is very cool.
4) The authors then hypothesize that human-generated data provides more information. They even train an informativeness prediction network to (re-)weight randomly generated examples which can then be used to train a reading comprehension model.

Pros of this work:
1) This work contributes a nice dataset that addresses a real problem faced by automatically generated datasets.
2) The breakdown of characteristics of questions is quite nice as well.
3) The paper is clear, well-written, and is easy to read.

Cons:
1) Overall, some of the claims made by the paper are not fully supported by the experiments. E.g., the paper claims that neural approaches are much worse than humans on CLOTH data -- however, they do not use state-of-the-art neural reading comprehension techniques but only a standard LSTM baseline. It might be the case that the best available neural techniques are still much worse than humans on CLOTH data, but that remains to be seen.
2) Informativeness prediction: The authors claim that the human-generated data provides more information than automatically/randomly generated data by showing that the models trained on the former achieve better performance than the latter on test data generated by humans. The claim here is problematic for two reasons:
   a) The notion of "informativeness" is not clearly defined. What does it mean here exactly?
   b) The claim does not seem fully justified by the experiments -- the results could just as well be explained by distributional mismatch without appealing to the amount of information per se. The authors should show comparisons when evaluating on randomly generated data.

Overall, this paper contributes a useful dataset; the analysis can be improved in some places.

---

> ### Author Response · Authors · 2018-01-05
> **Response**
>
> Thank you for your valuable review!
> 1. Please see our comment about the attention baseline in the top thread.
> 2. Indeed,  the statement about informativeness is not rigorous. With further experiments, we find that the results should be explained by a distributional mismatch instead of informativeness. Specifically, when the training set contains both the human-designed data and automatically generated data, the accuracy on automatically generated data increases if we have a higher proportion of automatically generated data in the training set. Please see Table 7 for more details. We restructured Section 4 and removed the informativeness section.
> 3. However, we believe human-designed data is a much better test bed for general cloze test with the following reasons: Human-designed data is different from automatically generated data since it leads to a larger gap between the model’s performance and the human performance. The model's performance and human's performance on the human-designed data are 0.484 and 0.860 respectively, leading to a gap of 0.376. The performance gap on the automatically-generated data is at most 0.185 since the model's performance reaches 0.815. Similarly, on Children’s Book Test where the questions are generated, the human performance is between 0.708 to 0.828 on four categories and the language model can nearly achieve human performance on the preposition and verb categories. Hence human-designed data is a good test base because of the larger gap between performances of the model and the human, although the distributional mismatch problem makes it difficult to be the best training source for out-of-domain cloze test such as automatically generated cloze test.

---

### Official Review · AnonReviewer1 · 2017-11-27
**Not convinced**

**Rating:** 4
**Confidence:** 4

**Review:**

1) this paper introduces a new cloze dataset, "CLOTH", which is designed by teachers. The authors claim that this cloze dataset is a more challenging dataset since CLOTH requires a deeper language understanding and wider attention span. I think this dataset is useful for demonstrating the robustness of current RC models. However, I still have the following questions which lead me to reject this paper.

2) I have the questions as follows:
i) The major flaw of this paper is about the baselines in experiments. I don't think the language model is a robust baseline for this paper.  When a wider span is used for selecting answers, the attention-based model should be a reasonable baseline instead of pure LM.
ii) the author also should provide the error rates for each kind of questions (grammar questions or long-term reasoning).
iii) the author claim that this CLOTH dataset requires wider span for getting the correct answer, however, there are only 22.4 of the entire data need long-term reasoning. More importantly, there are 26.5% questions are about grammar. These problems can be easily solved by LM.
iv) I would not consider 16% percent of accuracy is a "significant margin" between human and pure LM-based methods. LM-based methods should not be considered as RC model.
v) what kind accuracy is improved if you use 1-billion corpus trained LM? Are these improvements mostly in grammar? I did not see why larger training corpus for LM could help a lot about reasoning since reasoning is only related to question document.

---

> ### Author Response · Authors · 2018-01-05
> **Response**
>
> Thank you for your valuable review!
> i) Please see our comment about the attention baseline in the top thread.
> ii) The error rates for each kind of questions are added in Figure 1.
> iii) The questions in CLOTH dataset require a wider span when compared to automatically generated questions. We added more comparisons about human-designed data and automatically generated data in Section 4.1.
> iv) The margin 15.3% results from training on a large external dataset. Specifically, the 1-billion-word dataset is more than 40 times larger than our dataset. However, in practice, it requires too many computational resources to train models on such a large dataset. Hence, it is valuable to compare models that do not use external data. When we do not use external data, the margin between the best model and the human performance is 27.7%, which is still a large margin.
> v) Accuracies on all categories are improved if we train the LM on the 1-billion-word corpus. It shows that a large amount of data is necessary to learn complex language regularities. Please see Figure 1 for more details.

---

### Official Review · AnonReviewer2 · 2017-11-27
**This is an interesting dataset but the baselines are not very compelling.**

**Rating:** 4
**Confidence:** 4

**Review:**

This paper collects a cloze-style fill-in-the-missing-word dataset constructed manually by English teachers to test English proficiency.  Experiments are given which are claimed to show that  this dataset is difficult for machines relative to human performance.  The dataset seems interesting but I find the empirical evaluations unconvincing.  The models used to evaluate machine difficulty are basic language models.  The problems are multiple choice with at most four choices per question.  This allows multiple choice reading comprehension architectures to be used.   A window of words around the blank could be used as the "question".  A simple reading comprehension baseline is to encode the question (a window around the blank) and use the question vector to compute an attention over the passage.  One can then compute a question-specific representation of the passage and score each candidate answer by the inner product of the question-specific sentence representation and the vector representation of the candidate answer.  See "A thorough examination of the CNN/Daily Mail reading comprehension task" by Chen, Bolton and Manning.

---

> ### Author Response · Authors · 2018-01-05
> **Response**
>
> Thank you for your valuable review! Please see our comment about the attention baseline in the top thread.

---

### Author Response · Authors · 2018-01-05
**Attention Baselines**

Since all three reviewers suggested employing stronger baselines, specifically attention models, we will first clarify here:

1. We tested machine comprehension models (with attention) when we started working on the task but found that they do not significantly outperform the LSTM baseline. Specifically, the Stanford Attentive Reader achieves an accuracy of 0.487 on CLOTH while an LSTM based method has an accuracy of 0.484. We also implemented position-aware attention model [Zhang et al. 2017] to enable the model to use the distance information. It achieves an accuracy of 0.485. We have updated these results in the paper.
2. In fact, LSTM based language model is capable of modeling statistical regularities of language. Hill et al. 2015 show language models outperform memory networks and nearly achieves human performance on the verbs or prepositions questions of Children’s Book Test. A concurrent work also shows that language model is very good at modeling complex language regularities when trained on a large amount of data, although they use the LM to extract features instead of directly using it for prediction (Please see ICLR submission  “Deep contextualized word representations” ). Specifically, by replacing word vectors with hidden representations of LM, they achieve state-of-the-art results on six language tasks including textual entailment, question answering, semantic role labeling, coreference resolution, named entity extraction, sentiment analysis. Reasoning also benefits from LM features, e.g., the F1 on reading comprehension (SQuAD) improves from 81.1 to 85.3.
3. We hypothesize the attention models’ unexpected performance is due to the difficulty to learn to comprehend longer contexts when the majority of the training data only requires understanding short-term information. Specifically, there are 23.2% of questions that require a long-term context. Note that although the cloze test was previously introduced for evaluating reasoning abilities in the machine comprehension task, CLOTH does NOT focus on reasoning. We mentioned the difference in the related work section: “Our dataset focuses on evaluating language proficiency including knowledge in vocabulary, reasoning and grammar while the focus of reading comprehension is reasoning.” We have updated the paper to emphasize this point in the introduction.

Reference:
Zhang, Y., Zhong, V., Chen, D., Angeli, G., & Manning, C. D. (2017). Position-aware Attention and Supervised Data Improve Slot Filling. In Proceedings of the 2017 Conference on Empirical Methods in Natural Language Processing (pp. 35-45).
Hill, F., Bordes, A., Chopra, S., & Weston, J. (2015). The Goldilocks Principle: Reading Children's Books with Explicit Memory Representations. arXiv preprint arXiv:1511.02301.

---

### Decision · Program_Chairs · 2018-01-29
**ICLR 2018 Conference Acceptance Decision**

**Decision:**

Reject

**Comment:**

Meta score: 4

The paper presents a manually-constructed cloze-style fill-in-the-missing-word dataset, with baseline language modelling experiments that aim to show that  this dataset is difficult for machines relative to human performance.  The dataset is interesting but the fact that the experiments are confined to baseline language models
Pros:
 - interesting dataset
 - clear and well-written
 - attempt to move the field forward in an important area
Cons:
 - limited experimentation
 - language modelling approaches not appropriate baseline